# Heparin Blocks the Inhibition of Tissue Kallikrein 1 by Kallistatin through Electrostatic Repulsion

**DOI:** 10.3390/biom10060828

**Published:** 2020-05-28

**Authors:** Lina Ma, Jiawei Wu, Ying Zheng, Zimei Shu, Zhenquan Wei, Yinbiao Sun, Robin W. Carrell, Aiwu Zhou

**Affiliations:** 1Department of Pathophysiology, Shanghai Jiao Tong University School of Medicine, Shanghai 200025, China; malina-china@foxmail.com (L.M.); wujiaweiyes@live.cn (J.W.); 15805197619@163.com (Z.S.); weizhq@gmail.com (Z.W.); 2Randall Division of Cell & Molecular Biophysics, Faculty of Life Sciences & Medicine, King’s College London, New Hunt’s House, Guy’s Campus, London SE1 1UL, UK; yin-biao.sun@kcl.ac.uk; 3Department of Haematology, University of Cambridge, Cambridge CB2 0XY, UK; rwc1000@cam.ac.uk

**Keywords:** serpins, tissue kallikrein, heparin, serine protease, electrostatic repulsion

## Abstract

Kallistatin, also known as SERPINA4, has been implicated in the regulation of blood pressure and angiogenesis, due to its specific inhibition of tissue kallikrein 1 (KLK1) and/or by its heparin binding ability. The binding of heparin on kallistatin has been shown to block the inhibition of KLK1 by kallistatin but the detailed molecular mechanism underlying this blockade is unclear. Here we solved the crystal structures of human kallistatin and its complex with heparin at 1.9 and 1.8 Å resolution, respectively. The structures show that kallistatin has a conserved serpin fold and undergoes typical stressed-to-relaxed conformational changes upon reactive loop cleavage. Structural analysis and mutagenesis studies show that the heparin binding site of kallistatin is located on a surface with positive electrostatic potential near a unique protruded 3^10^ helix between helix H and strand 2 of β-sheet C. Heparin binding on this site would prevent KLK1 from docking onto kallistatin due to the electrostatic repulsion between heparin and the negatively charged surface of KLK1, thus blocking the inhibition of KLK1 by kallistatin. Replacement of the acidic exosite 1 residues of KLK1 with basic amino acids as in thrombin resulted in accelerated inhibition. Taken together, these data indicate that heparin controls the specificity of kallistatin, such that kinin generation by KLK1 within the microcirculation will be locally protected by the binding of kallistatin to the heparin-like glycosaminoglycans of the endothelium.

## 1. Introduction

Human kallistatin was first identified and characterized as a kallikrein binding protein [1,2]. Subsequent molecular cloning of its full-length cDNA showed that kallistatin is a glycoprotein with 401 residues and is expressed by many tissues such as liver, stomach, pancreas, kidney, and prostate [3]. Extensive studies have shown that kallistatin exerts a wide spectrum of biological activities in angiogenesis, apoptosis, and oxidative stress [4]. Kallistatin is a member of the serine protease inhibitor (serpin) superfamily and shares about 45% sequence identity with other serpins such as α_1_-antitrypsin and protein C inhibitor (PCI) [5,6]. Serpins share a conserved mouse-trap mechanism in inhibiting their target proteases [5,7]. The protease binds and attacks the scissile bond (P1-P1’, nomenclature of Schechter and Berger [8] in the reactive center loop of the native serpin (Appendix A). At the acyl-enzyme intermediate stage the serpin undergoes a rapid stressed-to-relaxed (S-to-R) conformational change where the reactive center loop is fully inserted into the central beta-sheet of the serpin resulting in translocation of the protease and distortion in the catalytic triad of the protease [7,9]. This leads to the formation of an SDS-stable covalent linkage and consequent inactivation of the protease complex [7]. Kallistatin was initially characterized as a specific inhibitor of tissue kallikrein 1 (KLK1), but it also inhibited other serine proteases such as KLK7 [10]. The protease specificity of a serpin is largely determined by its reactive loop sequence, especially the P1 residue, although in some cases, residues which do not belong to the reactive center loop can also critically affect specificity [9,11]. This appears to be so with kallistatin which unusually has a P2-P1-P1’ sequence of Phe-Phe-Ser targeting P1 Arg-preferred KLK1 [12,13]. 

Kallikreins are serine proteases which are widely expressed in various tissues and share the same conserved serine protease domain [14]. Human kallikreins can be divided into two categories: plasma kallikrein and tissue kallikreins (also called kallikrein-related peptidases, KLKs) [15,16]. Plasma kallikrein releases the vasoactive peptide bradykinin from high molecular weight kininogen [17]. KLKs have 15 members named from KLK1 to KLK15. In the human KLKs family, only KLK1 could release kinin from low molecular weight kininogen, while the other KLKs possess greatly reduced or no kininogen cleavage activity [16]. KLK1 is a key member of the kallikrein-kinin system that initiates and maintains vasodilation in mammalian system and its deficits are associated with cardiovascular and end-organ pathology [18]. KLK1 replenishment therapy is widely used in Asia for the treatment of diseases such as acute ischemic stroke [18].

Kallistatin belongs to a subclass of serpins known as the heparin binding serpins. In most cases, heparin binds to residues of the serpin helix D as with antithrombin [19,20,21,22,23], heparin cofactor II (HCII) [24,25,26], protease nexin 1 (PN1) [27,28], plasminogen activator inhibitor type 1 (PAI-1) [29], protein Z-dependent protease inhibitor (ZPI) [30,31], and lamprey angiotensinogen [32]. The binding of heparin promotes the interaction between serpin and protease either by an allosteric mechanism where heparin induces conformational changes in the serpin (Appendix A), or by a bridging mechanism where heparin links serpin and protease together forming a transient ternary complex [9,33,34] (Appendix A). However, it was proposed that the heparin biding site on kallistatin differed in being located near hH, similar to that of PCI [35,36]. It was also found that heparin blocked the interaction between kallistatin and KLK1, as compared to the acceleration of protease inhibition seen in other heparin binding serpins [37]. The detailed molecular mechanisms underlying these unique features of kallistatin are unclear. 

Here we prepared and crystalized recombinant kallistatin and solved the crystal structures of the reactive loop cleaved kallistatin and its complex with heparin. The detailed characterization of its properties in heparin binding and protease inhibition shows that the heparin binding site of kallistatin is located around a unique 3^10^ helix between helix H and strand 2 of β-sheet C of kallistatin, and that heparin binding on kallistatin blocks the docking of KLK1 onto kallistatin through electrostatic repulsion.

## 2. Materials and Methods 

### 2.1. Materials

The cDNA for human kallistatin, human kallikrein-related peptidase 1 (KLK1) was purchased from GENECHEM. Fluorescent substrate H-D-Val-Leu-Arg-AMC for KLK1 was synthesized by BankPeptide. Low molecular weight (LMW) heparin with an average molecular weight of ≈5 kDa and high molecular weight (HMW) heparin with an average molecular weight of 17–19 kDa were purchased from Sigma. Heparin oligosaccharides with 4, 6, 8, 10, and 12 saccharide units (DP4, DP6, DP8, DP10, and DP12) were from Iduron (Manchester, UK). Columns for protein purification were purchased from GE Healthcare. All reagents and kits for crystallization were purchased from Hampton Research. Precast 12% SDS-PAGE gels were purchased from Genscript (Nanjing, China) and run at 150 volts for ≈ 45 min using MOPS buffer. Protein samples were treated with reducing SDS loading buffer at 95 °C for 5 min before electrophoresis.

### 2.2. Cloning, Mutagenesis, Expression, and Purification of Recombinant Proteins

Recombinant kallistatin was prepared from *Escherichia coli* as previously described [38]. Briefly, the coding sequence covering residues 45–427 of human kallistatin was cloned in *E. coli* expression vector pE-SUMO3. All mutants were constructed by PCR mutagenesis using the KOD Plus mutagenesis kit (TYOBO). The expression plasmid construct was transformed into BL21 (DE3) cells and grown in 2 × YT at 37 °C until optical density at 600 nm reached 0.8. Then isopropyl-β-D-1-thiogalactopyranoside (IPTG) was added to a final concentration of 0.2 mM, and the culture was transferred to a 20 °C shaker for a further 12 h. The cells were collected by centrifugation and resuspended in ice cold buffer A (20 mM Tris-HCl, pH 7.4, 0.5 M NaCl, and 20 mM imidazole) and disrupted by a high-pressure cell breaker. The supernatant of the cell lysate was loaded onto a 5 mL HisTrap FF crude column and eluted by a 0.02–0.2 M imidazole gradient. The peak fractions were collected. Then the SUMO3-fusion protein was digested by protease SENP2 to release the SUMO3 tag. Kallistatin was further purified by a 5 mL HiTrap SP column and eluted by a NaCl gradient (0.05–1 M in 10 mM MES pH 6.0). Notably, during storage or crystallization, recombinant kallistatin was cleaved between P2 and P1 residues by an unknown protease and migrated as a doublet band on SDS-PAGE.

Recombinant tissue kallikrein KLK1, and its variants KLK1-99Lm, where nine residues from its 99 loop were deleted, and KLK1-70Lm with residues D75, D76 and E77 replaced by arginines, were prepared from *E. coli* expression system through refolding [39,40]. The sequence coding resides 25-262 were cloned in expression vector pE-SUMO3. KLK-70Lm and KLK1-99Lm mutants were constructed by PCR mutagenesis using the KOD Plus mutagenesis kit (TYOBO). Expression plasmids were transformed into BL21 (DE3) cells and grown in 2 × YT at 37 °C until optical density at 600 nm reached 0.8. Then IPTG was added to a final concentration of 0.2 mM and shaker for a further 12 h. The cells were collected by centrifugation and resuspended in 20 mM Tris-HCl, pH 7.4, 1% (v/v) Triton X-100, 20 mM EDTA and disrupted by a high-pressure cell breaker. The cell lysate was then centrifuged and the pellet containing inclusion bodies was collected and washed twice with 20 mM Tris-HCl, pH 7.4, 20 mM EDTA, and 1.0 M NaCl. Finally, the inclusion bodies were washed with 20 mM Tris-HCl, pH 7.4, 20 mM EDTA and dissolved in 6.0 M guanidinium chloride, 20 mM Tris-HCl, pH 8.0, 0.5 mM EDTA, and 10 mM DTT. Protein refolding was initiated by rapid dilution of dissolved inclusion bodies into refolding buffer containing 50 mM Tris-HCl, pH 8.0, 0.6 M arginine, 20 mM CaCl_2_, 10% (v/v) glycerol, 10 mM reduced glutathione, and 1 mM oxidized glutathione with a final protein concentration less than 0.3 mg/mL. The refolding solution was kept at 4 °C for 16 h and then dialyzed against 20 mM Tris-HCl, pH 7.4, 0.5 M NaCl, and 20 mM imidazole. The refolded KLK1 fusion protein was purified by a 5 mL HiTrap FF column. KLK1 was activated by protease SENP2 which releases the SUMO3 tag from the fusion protein. Activated KLK1 was loaded onto a 5 mL Hitrap SP HP column and eluted by a NaCl gradient (0.05–1 M). The peak fractions were collected, concentrated, and buffer exchanged into 10 mM Tris-HCl, pH 7.4, 0.1 M NaCl. 

### 2.3. Crystallization, Data Collection, and Structure Determination

The initial conditions for crystallization were screened at 22 °C by the sitting-drop vapor-diffusion method using screening kits from Hampton Research in MRC2 plates. Crystals were initially grown from a mixture of 200 nL protein solution (10 mg/mL in 10 mM Tris-HCl, pH 7.4, 0.1 M NaCl) and 200 nL precipitant solution (40% tert-butanol) equilibrated against 80 μL reservoir solution. A cryosolution consisting of 10% glycerol and 10% ethylene glycol was directly added to the drop and the crystals were quickly picked up using a micro-loop and flash-cooled in liquid nitrogen. Owing to rapid evaporation of *tert*-butanol, the crystals float around in the drop and crack once the well is opened. Therefore, for the heparin soaking experiment, 100 µM heparin pentasaccharide in the cryo-solution was similarly directly added to the drop and a small piece of cracked crystal was immediately picked up using a micro-loop and flash-cooled in liquid nitrogen. This synthetic heparin pentasaccharide was used previously in obtaining antithrombin-heparin crystal structure [22]. Diffraction data were collected on beamline BL17U at SSRF (Shanghai, China). The data set was indexed and processed with iMosflm [41] and scaled with AIMLESS from the CCP4 suite [42,43] A random choice of 5% of the data was selected as the R-free set. The structures were solved by molecular replacement with Phaser [44] using reactive loop cleaved PCI structure (PDB 3DY0) as search model. Notably, a phenix polder map for the kallistatin-heaprin complex structure was calculated (Appendix A). This confirmed that heparin is present in the crystal structure and the ligand was refine to have a partial occupancy of 0.69 by phenix.refine. The structure was built in COOT [45] and refined using Refmac and Phenix.refine [46] to good geometry (Table 1). Figures were produced with PyMOL software. The surface electrostatic representation was analyzed by APBS (Adaptive Poisson-Boltzmann Solver) [47].

### 2.4. Stoichiometries of Inhibition and Rates of Inhibition

Stoichiometries of inhibition (SI) were measured using similar procedures as previously described [32] by incubating KLK1, KLK1-70Lm, and KLK1-90Lm (1 μM) with increasing concentrations of kallistatin variants in phosphate buffered saline (PBS) with 0.1% PEG 8000 for 2 h at 37 °C. Residual protease activity was determined by diluting the reaction mixture into 0.2 mM H-D-Val-Leu-Arg-AMC for KLK1. Linear regression analysis of the decrease in protease activity with an increasing concentration of kallistatin yielded the estimates for the stoichiometry of inhibition as the intercept on the abscissa. Each measurement was repeated three times with the mean SI value ± SD calculated. 

The rates of inhibition by recombinant kallistatin variants were determined at room temperature (22 ± 1 °C) by a discontinuous assay procedure as previously described [32]. Briefly, under pseudo–first-order conditions, 10 μL of 10 nM proteases was mixed with 10 μL of 100–500 nM kallistatin variants in PBS with 0.1% PEG 8000. The residual protease activity was determined at timed intervals by diluting the reaction mixture into 200 μL of the assay buffer containing substrate. The observed rate constant, *k*_obs_, for the reaction was obtained from the slope of a semilog plot of the residual protease activity against time, and the apparent second-order rate constant, *k*_app_, was calculated by dividing *k*_obs_ with the initial serpin concentration. The measurement for each kallistatin variant was repeated three times with the mean *k*_app_ value ± SD calculated. The product of *k*_app_ × SI represents the second order rate constant (*k*_2_).

The effect of low molecular weight heparin (LMWH) on the interactions between kallistatin and KLK1-70Lm was similarly assessed as previously described [10,48]. A total of 1 µM rKAL was incubated with different concentrations of LMWH (0, 0.005, 0.025, 0.05, 0.25, 0.5, 1, and 2.5 mg/mL) for 5 min at room temperature, and then admixed with 10 nM KLK1-70Lm and the residual KLK1-70Lm activity was measured using 0.2 mM fluorescent substrate H-D-Val-Leu-Arg-AMC at different time intervals. The observed rate constants, *k*_obs_, were then plotted with the concentrations of LMWH. 

### 2.5. Heparin Affinity Chromatography of Kallistatin Variants

The relative heparin binding affinities of recombinant kallistatin variants were assessed by a heparin-Sepharose column [32]. About 200–400 µg of protein was loaded onto a 1 mL HiTrap heparin column pre-equilibrated with 10 mM Tris-HCl, pH 7.4, and eluted with a 20-column volume of 0–500 mM NaCl gradient using the AKTA system (GE Healthcare). The protein absorbance at 280 nm was recorded with the salt concentration of peak identified and the fractions analyzed by SDS-PAGE.

### 2.6. Model of the Kallistatin-KLK1-Heparin Ternary Complex

There are about a dozen different serpin-proteases Michaelis complex crystal structures in the Protein Data Bank. Although the relative positions of serpin and protease are somewhat different, it is apparent that all the proteases docked on top of the serpin are very close to the hH of the corresponding serpins. To illustrate the relative positions of KLK1 and kallistatin in their Michaelis complex without extensive optimization of the surface loop conformations of both proteins, a model of KLK1 and kallistatin Michaelis complex was prepared from the thrombin/antithrombin complex structure (PDB 1SR5) with KLK1 and kallistatin superposed onto thrombin and antithrombin, respectively. This resulted in minimum clashes between the two molecules. The heparin tetra-saccharide was docked onto kallistatin using ClusPro web server [49] and one of the top five output coordinates with the heparin chain in an optimal orientation forming interactions with key residues R259, R300, K307, and R394 was selected and further extended for presentation.

## 3. Results

### 3.1. The Overall Structures of Kallistatin and Its Complex with Heparin

Recombinant human kallistatin (rKAL) was prepared from an *Escherichia coli* expression system. SDS-PAGE was used to visualize the purity of rKAL and its interaction with proteases. As shown in Figure 1, the purified rKAL readily formed SDS-stable covalently linked rKAL-KLK1 complex with a molecular weight of ≈ 70 kDa, however in the presence of low molecular weight heparin (LMWH), little complex could be seen from the gel (Figure 1A). The closely migrated doublet bands of kallistatin in lane 2 reflects some of the rKAL being cleaved by KLK1 during inhibition, which is consistent with the stoichiometry of inhibition (SI) measurement (Table 2). The inhibition of KLK1 by rKAL was also quantitatively assessed by an enzymatic assay using a KLK1’s fluorescent substrate. rKAL almost completely inhibited the KLK1 activity in the absence of heparin, but this inhibition is significantly reduced by the presence of 0.5 mg/mL of high or low molecular weight heparins with ≈ 60%–80% of KLK1’s activity remaining after 30 min incubation (Figure 1B). Notably, as the formation of the covalently linked rKAL-KLK1 complex is irreversible and heparin binding of rKAL is reversible, prolonged incubation of rKAL and KLK1 even in the presence of high concentrations of heparin would lead to a gradual increase of rKAL-KLK1 complex formation and the loss of KLK1 activity. Subsequent second-order association rate constant measurement showed that rKAL inhibited KLK1 with a *k*_2_ of 3.4 × 10^3^ M^−1^ s^−1^ (Table 2). These results are consistent with previous findings that kallistatin is an inhibitor of KLK1 and their interaction can be blocked by heparin [10,48]. 

To further characterize kallistatin, we firstly crystallized recombinant kallistatin and then soaked its crystals with a synthetic heparin pentasaccharide [22]. The crystal structures of rKAL and its complex with heparin pentasaccharide were then solved at 1.9 and 1.8 Å resolution, respectively (Table 1), using reactive loop cleaved PCI (PDB 3DY0) as the searching model [35]. The N-terminal amino acid sequencing of kallistatin from the crystallization drop indicated that it was cleaved between residues F387 and F388 (P2-P1 position) of the reactive center loop by an unknown protease during crystallization, which commonly occurs with many serpins [50]. One copy of reactive loop cleaved kallistatin was found in the asymmetric unit. Two heparin saccharide units were built in the rKAL-heparin structure. No significant structural difference in kallistatin was apparent between the two crystal structures. Overall, kallistatin showed a typical serpin fold in a relaxed conformation, with the cleaved reactive center loop completely inserted into the central β-sheet A as a middle strand (Figure 1C). When compared with other relaxed serpin structures, such as cleaved α_1_-antitrypsin (PDB 3NDD), or cleaved PCI (PDB 3DY0), the root-mean-square-deviations of atomic positions were less than 1.5 Å indicating a remarkably conserved tertiary structure. Nevertheless, there are some differences. Sequence alignment of kallistatin with other human serpins (Figure 1E) indicated that kallistatin has a unique four-residue insertion (R^306^KRN^309^) between helix H and strand 2 of β-sheet C (s2C). In the kallistatin crystal structure, this four-residue insertion forms an extra 3^10^ helix (termed hH’) and it is packed against a bulky W254 from what is the hormone binding pocket in some other serpins [51]. The sidechain of W254 is further stabilized by cation-Pi and Pi-Pi interactions from nearby residues R267, F277, and Y311 (Figure 1D). R267 also forms salt bridges with D269 of s5B. Therefore, the tight packing within the ‘hormone binding pocket’ of kallistatin effectively forces the inserted four residues to stay as a protruded 3^10^ helix on the top of kallistatin. There are three potential glycosylation sites in kallistatin (Appendix A), which are far away from the protease docking position on kallistatin.

### 3.2. Heparin Binding Site of Kallistatin

In the crystal structure of kallistatin complexed with heparin pentasaccharide, there is clear electron density for two terminal saccharide units G and H with four sulfonate groups (Figure 2A). A phenix polder map calculated for the ligand further supported the presence of heparin pentasaccharide in the crystal structure. The stereoview of the electron density map around the heparin saccharides shows the detailed interactions of both saccharides (Figure 2B). The ring of saccharide unit H is sandwiched by the sidechains of R259 and Y260 with 6-O-sulfonate dipping into a shallow surface pocket (Figure 2B,C). R259 of s6B forms three salt bridges with two sulfonate groups from the two saccharides, and R300 of hH forms two salt bridges with the 6-O-sulfonate of saccharide unit H. The fourth sulphate, 3-O-sulfonate of saccharide H, has no direct interaction with kallistatin. The other three saccharides including saccharide unit F which harbored the key 3-O-sulfonate critical for antithrombin activation [40] were not built due to poor electron density. Notably residues K307 and R308 are involved in crystal packing which likely hinders direct heparin binding to these residues during soaking. Similarly, the soaked heparin pentasaccharide forms salt bridges with K150 of a symmetry-related kallistatin molecule. As K150 is located between hE and s2A at the bottom of kallistatin, this interaction is unlikely to play a significant role in the heparin binding of kallistatin. 

To further elucidate the property of this area, we identified all the basic residues around the heparin saccharide binding site (Figure 2D) and calculated the surface electrostatic potentials of kallistatin (Figure 2E). This clearly showed that kallistatin has an area with extensive positive electrostatic potential around the 3^10^ helix involving residues K307, R308, K312, and K313 as proposed previously [37], and other residues from helix H (R300), s6B (R259), and s1C (R394 and H395). Overall, the kallistatin-heparin complex structure shown here indicates that the surface with positive electrostatic potential is the likely heparin binding site of kallistatin.

PCI is another heparin binding serpin with a heparin binding site located close to helix H [35]. R229 of PCI, which corresponds to R259 in kallistatin, has also been implicated in heparin mediated inhibition of activated protein C (APC) by PCI (Figure 2F). However, there are differences in the exact locations of the heparin binding sites on these two serpins. The heparin binding site of PCI is located near the lower half of hH, while the heparin binding site of kallistatin is near the top of hH (Figure 2E,F). There are also minor changes in the positions of helix H and G when superposed (Figure 2G).

To elucidate the contribution of this surface with positive electrostatic potential of kallistatin towards heparin binding, we systematically mutated the positively charged residues from this area to Asp individually and assessed the heparin-binding affinities of these mutants by a heparin-Sepharose column (Figure 3). Wild type kallistatin was eluted from the heparin-Sepharose column at about 300 mM NaCl, while R259D, R308D, and R394D mutants were eluted earlier at ≈160 mM NaCl indicating substantial decrease in their heparin-binding affinity (Figure 3). Other mutations on R300, K307, K312, K313, and H395 also resulted in decreased heparin-binding. The binding affinity of the double mutant of 307/308D or 312/313D decreased even further with these two mutants being eluted from the heparin column at 95 and 125 mM NaCl, respectively (Figure 3), which is consistent with previous findings [37]. Therefore, this confirms that the surface positive electrostatic potential area is the heparin binding site of kallistatin.

We also tested if these mutations on kallistatin would affect the inhibitory activity of kallistatin towards KLK1 by measuring the association rate constants of these mutants (Table 2). This showed that the inhibition of KLK1 by rKAL-R394D, rKAL-K312/313D, and rKAL-K307/308D mutants decreased only by about 2-fold. Other residues such as R259 and R300 seemed to have a very limited effect on the KLK1 inhibition by kallistatin. This indicates that the heparin binding site residues of kallistatin are involved in KLK1 inhibition possibly through exosite interactions between the bodies of the KLK1 and kallikrein, but play a relatively minor role. 

### 3.3. The Mechanism of Heparin in Blocking KLK1/Kallistatin Interaction

In general, heparin is considered as a cofactor in promoting the inhibition by serpins of their target proteases, however it functions as a blocker in the interaction between kallistatin and KLK1 (Figure 1). KLK1 shares a typical serine protease domain with other members of the KLKs family, but its kallikrein loop (also called 99 loop) is about nine residues longer than those of other KLKs (Figure 4A). During KLK1 inhibition by kallistatin, it is expected that the reactive center loop of kallistatin will dock into the substrate binding pocket of KLK1 forming the initial encounter complex, also called Michaelis complex as seen in many serpin-protease Michaelis complex crystal structures [34,51,52]. There are about eight serpin-protease Michaelis complex structures deposited in Protein Data Bank from the five serpins listed in Figure 1E [21,34,51,52,53,54]. The relative positions of protease and serpin amongst these complexes are slightly different and the proteases often utilize different surface loops to form different exosite interactions with different parts of the serpin molecules [34]. As the reactive loop of kallistatin has the same length as those of antithrombin and PCI (Figure 1E), two different antithrombin-thrombin-heparin ternary Michaelis complex structures (PDB 1SR5 and 1TB6) reflecting different stages of the Michaelis complex formation between antithrombin and thrombin [21,54], and one PCI-thrombin complex structure [55] were selected for initial evaluation. Superposition of these structures together (Appendix A) showed that thrombin docked differently on top of the serpins forming different exosite interactions. Molecular modelling of the KLK1-kallistatin Michaelis complex by substitution of the protease and serpin in the structure with KLK1 and kallistatin, respectively, showed that minimum clashes between the bodies of the protease and serpin would occur if based on PDB 1S5R. Notably, kallistatin has similarly positively charged residues at P5’ and P10’ positions of the reactive loop as those antithrombin. Therefore, the model of kallistatin and KLK1 Michaelis complex was subsequently generated by simply overlaying KLK1 and kallistatin onto the antithrombin-thrombin complex structure (PDB 1SR5) to indicate the relative positions of KLK1 and kallistatin in the complex [21].

This model showed that the kallikrein loop of KLK1 would be close to the heparin binding site of kallistatin (Figure 4B). It would be plausible that heparin binding on this area would block the docking of KLK1 onto kallistatin due to this long kallikrein loop. To test this, we made a KLK1 mutant (KLK1-99Lm) where this loop is shortened by nine residues to the length of that of KLK7 and assessed the effect of heparin on its inhibition by kallistatin. This mutant formed complexes with kallistatin slowly (Figure 4A, lane 2) and the second-order association rate constant of this reaction is only ≈ 200 M^−1^s^−1^. This confirms that the kallikrein loop is involved in forming exosite interactions with the body of kallistatin. However, to our surprise, the inhibition of this mutant by kallistatin remained blocked by the presence of heparin (Figure 4A, lanes 3 and 4). This indicates that heparin mediated inhibition of KLK1-kallistatin interaction is not due to the longer kallikrein loop of KLK1.

Subsequent molecular surface electrostatic potential analysis shows that KLK1 is an acidic protease with a highly negatively charged surface (Figure 4C). It is apparent that in the Michaelis complex of kallistatin with KLK1 this negatively charged surface is spatially very close to the positively charged heparin binding site of kallistatin. Any extension of the heparin chain bound on kallistatin, modeled with six heparin saccharides in Figure 4C would potentially cause electrostatic repulsion between the negative charged KLK1 surface and the heparin chain, which is also negatively charged. To further confirm this, we tested on SDS-PAGE the blocking effect of heparin oligosaccharides of different length. As seen in Figure 4D, rKAL readily formed a complex with KLK1 in the absence of heparin (lane 2) and little blocking effect was seen with 1 mM heparin containing four saccharides (DP4, lane 3). A substantial reduction of complex formation was observed with 1 mM DP6 (lane 4) while a near complete loss of complex formation was observed in the presence of 1 mM DP8 or DP10 or DP12 (Figure 4D, lanes 5–7). This further emphasizes that the heparin binding site of kallistatin is within close proximity to the docked KLK1 in the Michaelis complex, which makes this heparin mediated blocking possible. As 3-*O*-sulfonate is very rare in these normal heparin oligosaccharides and it is not involved in forming interactions with kallistatin in the crystal structure (Figure 2), it is unlikely that it would play an essential role in blocking the interaction between KLK1 and kallistatin.

As the exosite 1 of KLK1, the so called 70–80 loop of the serine protease, is likely to interact with the extension of a linear heparin chain bound on kallistatin, here we mutated the negatively charged residues of the exosite with basic residues present in thrombin and assessed the inhibition of this mutant (KLK1-70Lm) by kallistatin. As shown in Figure 4A (lane 5), the mutant formed SDS-stable complexes with kallistatin in the absence of heparin, but in contrast to wild type KLK1, more complexes between kallistatin and KLK1-70Lm were formed in the presence of heparin (Figure 4A, lanes 6 and 7). The second-order rate constant measurement showed that rKAL inhibited KLK1-70Lm with a *k*_2_ of 1.8 × 10^3^ M^−1^ s^−1^, similar to that of wild type KLK1 (Appendix A), but its reaction with rKAL could be accelerated by ≈ 5-fold in a heparin dose-dependent manner (Figure 4E). This indicates that heparin could function through a similar bridging mechanism in promoting inhibition of KLK1-70Lm by kallistatin as seen in heparin mediated thrombin inhibition by many other serpins [34]. 

More informatively, the existence of a ternary complex of kallistatin, KLK1-70Lm and heparin implies that the heparin binding site of kallistatin could engage heparin and form exosite interactions with KLK1 simultaneously. This is consistent with the observation that DP4 binding does not affect the interaction between KLK1 and kallistatin (Figure 4D). It is plausible that heparin binds across the center of the heparin binding area of kallistatin while KLK1 surface loops form exosite interactions with the rim of this area involving residues such as K307, R308, K312, and K313 (Table 2). Additionally, this finding directly excludes the possibility that heparin bound on kallistatin would inhibit KLK1-kallistatin interaction through preventing the exosite interactions between KLK1 and kallistatin. Therefore, heparin blocks the inhibition of KLK1 by kallistatin most likely through preventing KLK1 from docking onto kallistatin due to the electrostatic repulsion between the long heparin chain and the exosite 1 of KLK1. Conversely, heparin could promote the inhibition of proteases with positively charged exosite 1 such as KLK1-70Lm (Figure 4A, lanes 6 and 7), or thrombin by kallistatin with its P1 residue mutated to Arg as shown previously [12].

## 4. Discussion

Serpins inhibit their target proteases mainly through a two-step process (Figure 5A). The reactive center loop of the serpin is firstly docked into the substrate binding pocket of the protease forming a Michaelis complex which is reversible. The second step, which is irreversible, involves cleavage of the reactive loop and subsequent dramatic serpin conformational changes leading to the formation of a covalently linked serpin-protease complex and inactivation of the protease. The specificity of serpins is primarily determined by the P1 and immediately adjacent residues. However, serpins have also adopted the use of cofactors such as heparin, vitronectin, and protein Z etc., to further refine their overall specificity and activity mainly through optimization of the Michaelis complex formation [9,34]. Heparin binds to helix D of antithrombin, heparin cofactor II, PAI-1 or PN1, and promotes the inhibition of target proteases either through an allosteric mechanism as seen in FXa inhibition by antithrombin where heparin induces an optimal conformation of antithrombin [22], or a structurally well-defined bridging mechanism as seen in the heparin medicated inhibition of thrombin by antithrombin where heparin links the serpin and thrombin simultaneously [34]. Interestingly, it has been shown that heparin binding on kallistatin and PCI would block their inhibition of KLK1, but promote their inhibition of other serine proteases such as APC and thrombin [12,52]. The heparin binding site on PCI has been elucidated structurally and heparin promotes the inhibition of thrombin or APC through the bridging mechanism where heparin binds hH of PCI and exosite 2 of thrombin or exosite 1 of APC [35]. However, it remains unclear how heparin would block the inhibition of KLK1 by these two serpins.

Here we solved crystal structures of the reactive loop cleaved kallistatin and its complex with heparin and revealed that the heparin binding site of kallistatin is located around a four-residue insertion (R^306^KRN^309^) between hH and s2C. As the heparin pentasaccharide was soaked into the crystal of kallistatin and some surface residues of kallistain are involved in crystal packing, the heparin binding orientation on kallistatin in solution is likely to be different and there could even be alternative ways for heparin to bind the same serpin as shown previously with PCI [35]. Mutagenesis study showed that the heparin binding site involved key residues such as R259, R300, R308, and R394 of kallistatin, which mirrored a similar heparin binding surface on PCI. This indicated that heparin bound on the similar areas of these two serpins could block the inhibition of KLK1 through a similar molecular mechanism. We tested the role of the heparin binding residues of kallistatin in KLK1 inhibition. There was only a minor decrease in association rate (≈ 2-fold) between kallistatin mutants (rKAL-312/313D) and KLK1(Table 2). KLK1 inhibition by this mutant could still be blocked at higher concentrations of heparin (data not shown). Similarly, shortening the long kallikrein loop of KLK1 did not affect the blocking function of heparin either (Figure 4A). This indicated that the blocking effect of heparin on KLK1 inhibition by kallistatin was not due to steric hindrance of the exosite interactions between KLK1 and kallistatin. This finding is in line with previous studies on PCI-KLK1 interactions which excluded the direct involvement of basic amino acid residues present in the heparin biding site of PCI on the KLK1-PCI interaction [52].

The method or manner by which heparin blocks the inhibition of KLK1 by kallistatin became apparent during molecular modelling of the Michaelis complex of kallistatin with KLK1. The surface electrostatic calculation revealed that KLK1 has a molecular surface with a high negative electrostatic potential. When KLK1 forms a Michaelis complex with kallistatin, in which the reactive center loop of kallistatin is inserted in the active site pocket of KLK1, the heparin binding surface of kallistatin with positive electrostatic potential is spatially very close to KLK1. If heparin were bound on kallistatin before the formation of the Michaelis complex, most plausibly, it would prevent KLK1 from docking onto kallistatin due to electrostatic repulsion between negatively charged heparin saccharides and KLK1 (Figure 5B). It is highly likely that heparin blocks the inhibition of KLK1 by PCI through the similar mechanism. This blockade is largely due to the unique heparin binding sites around hH on kallistatin and PCI, which are very close to where proteases would dock, and the unusual surface property of KLK1 amongst serine proteases. Our subsequent charge-reversal mutagenesis study on the exosite 1 of KLK1 further established that heparin could promote the inhibition of KLK1-70Lm by kallistatin (Figure 4E), reminiscent of the heparin mediated accelerated protease-inhibition by other serpins [34] (Appendix A). Notably kallistatin is also an efficient inhibitor of KLK7 [10], which has many positively charged residues in its exosite 1, it is plausible that heparin could accelerate the inhibition of KLK7 by kallistatin through a bridging mechanism. Intriguingly, vaspin/SERPINA12 inhibits both KLK7 and KLK14, but heparin accelerates its inhibition of KLK7, but prevents the inhibition of KLK14 [53,54]. It is unclear if this blockade of KLK14 inhibition is through a similar electrostatic expulsion mechanism shown here.

Serpin cofactors, such as heparin, vitronectin or protein Z, not only regulate the inhibitory activity or specificity of the serpin, but also direct it to its target tissues to perform its functions in a localized environment. For example, heparin-like glycosaminoglycans located on extracellular matrix and the surface of vascular smooth muscle cells and endothelial cells will facilitate protease inhibition by bringing the serpin into close proximity to the protease generation. Interestingly, kallistatin has been shown to compete with VEGF in binding to heparin-like glycosaminoglycans in extracellular matrix and to consequently modulate angiogenesis [55]. As shown here, even though KLK1 is the putative in vivo target protease of kallistatin, inhibition is unlikely to occur when the kallistatin is bound to the heparinoids of the endothelium (Figure 5B). This localized inactivation of kallistatin will facilitate kinin generation by KLK1 and the subsequent activation of kinin-receptor on the cell surface—in keeping with the function of kallistatin as a vasodilator [56].

## 5. Conclusions

Our study here revealed that the unique heparin binding site of kallistatin is located around a protruded 3^10^ helix near helix H and that heparin binding controls the activity of kallistatin by blocking its inhibition of KLK1 through electrostatic repulsion between heparin and the negatively charged surface of KLK1. 

## Figures and Tables

**Figure 1 biomolecules-10-00828-f001:**
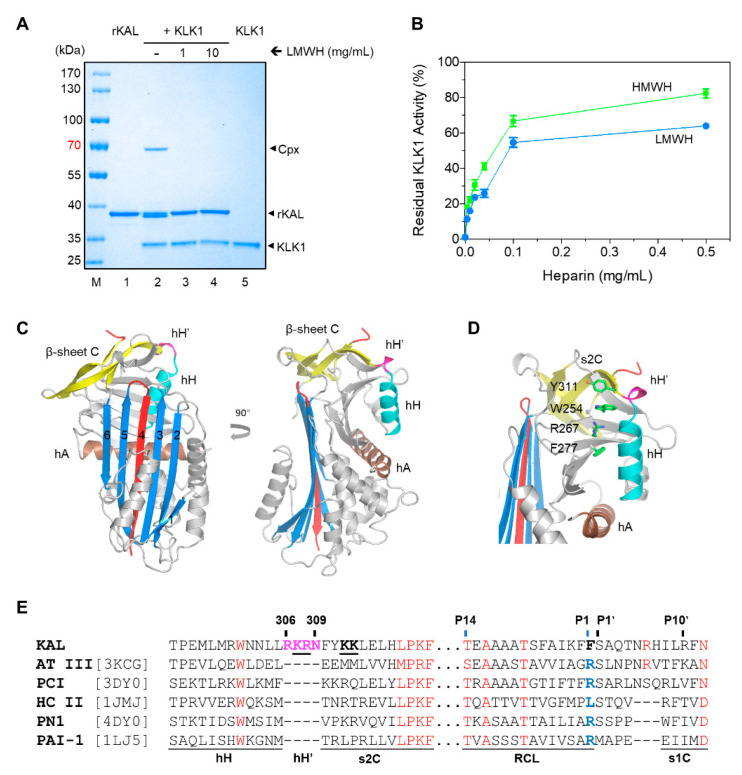
Characterization of recombinant Kallistatin. (**A**) A total of 2 µg of rKAL was incubated with 1 µg tissue kallikrein 1 (KLK1) for 2 min at room temperature in the absence or presence of LMWH and then analyzed by reducing SDS-PAGE. M, molecular weight marker. (**B**) Inhibition of KLK1 was also assessed by an activity assay where 500 nM of rKAL was mixed with different concentrations (0, 0.004, 0.01, 0.02, 0.04, 0.1, 0.5 mg/mL) of LMW (blue) or HMW (green) heparin and then incubated with 10 nM KLK1 at 37 °C for 30 min. The remaining activity of KLK1 was then measured using 0.2 mM fluorescent substrate H-D-Val-Leu-Arg-AMC. The normalized residual KLK1 activity (±SD, *n* = 3) was plotted against the concentration of heparin. (**C**) The structure of reactive loop cleaved kallistatin. The cleaved reactive center loop (RCL) (red) is completely inserted into the central β-sheet A (blue) as a middle strand. Residues R^306^KRN^309^ between helix H (cyan) and strand 2 of β-sheet C (yellow) form an extra 3^10^ helix (hH’, pink). (**D**) W254 from the ‘hormone-binding pocket’ is stabilized by R267, F277, and Y311. (**E**) Partial amino acid sequence alignment of kallistatin with other heparin binding serpins. The unique four-residue insertion is located between hH and strand 2 of β-sheet C (s2C). The RCL of kallistatin has the same length as those of antithrombin (ATIII) and protein C inhibitor (PCI), but it is four residues longer than those of HCII, PN1, and PAI-1. Highly conserved residues are shown in red and residues R^306^KRN^309^ of hH’ are shown in pink. Key residues K307, R308, K312, and K313 involved in heparin binding are underlined.

**Figure 2 biomolecules-10-00828-f002:**
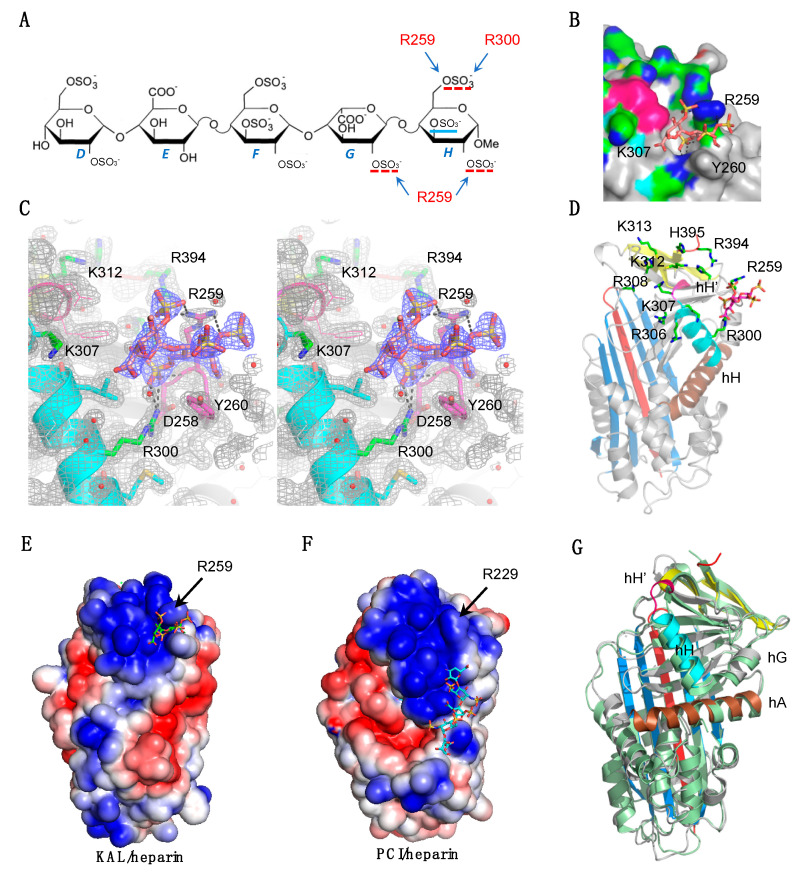
Crystal structure of kallistatin complexed with heparin. (**A**) Diagram showing the structure of heparin pentasaccharide. Three sulfonate groups (highlighted with red underlines) from saccharide unit *G* and *H* formed key interactions with R259 and R300 of kallistatin while the 3-*O*-sulfonate group (blue underline) from saccharide unit *H* has no direct interaction with the protein. (**B**) Stereoview of the electron density map covering the heparin saccharides and surrounding area with saccharide *G* and *H* built in the electron density. The 2F_0_-F_c_ electron density is contoured at 1σ. R300, stabilized by ionic interactions with D258, forms two salt bridges with 6-*O*-sulfonate from saccharide unit *H*. R259 forms two salt bridges with 2-*O*-sulfonate group of saccharide *G* and one salt bridge with 2-*O*-sulfonate group from saccharide *H*. Notably, 6-*O*-sulfonate from saccharide *H* also forms a hydrogen bond with the main chain amino group of R259. (**C**) Surface presentation showing saccharide unit *H* (sticks) is sandwiched between R259 and Y260 with its 6-*O*-sulfonate dipping into a shallow pocket forming interactions with R300 and R259 as seen above. The 3^10^ helix hH’ is colored pink and carbon atoms from surrounding residues are coloured in green with nitrogen atoms in blue. (**D**) The positively charged residues R259, R300, R306, K307, R308, K312, K313, R394, and H395 (green) near hH’ and the heparin saccharides (pink) are shown in sticks. (**E**) Surface electrostatic analysis shows that kallistatin has an area with extensive positive electrostatic potential on top of hH involving surrounding residues from hH’. (**F**) The heparin binding site of PCI (PDB 3DYO) is located to the lower part of hH with the extension of the heparin pentasaccharide expected to form interaction with R229. The orientations of KAL in (**D**,**E**) are same as PCI in (**F**). The electrostatic surface potential was calculated by APBS (red is negative and blue is positive, −3 to +3 kT/e). (**G**) The overlaid structures of PCI and rKAL showing an overall similar configuration. Helices G and H have slightly different positions and orientations. PCI is colored light green and kallistatin is color coded as in Figure 1C.

**Figure 3 biomolecules-10-00828-f003:**
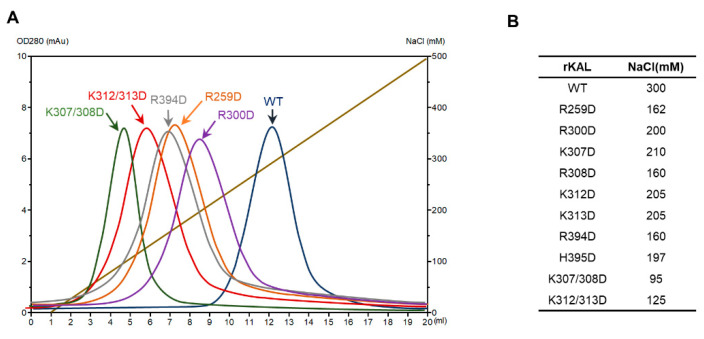
Heparin binding affinity of kallistatin variants assessed by a heparin-Sepharose column. (**A**) Kallistatin variants were loaded onto a heparin-Sepharose column (1 mL bed volume) and then eluted with a gradient of NaCl solution. The elution profile is shown in (**A**) with the absorbances at 280 nm wavelength and salt gradient as Y axes. (**B**) The peak positions identified from the absorbance were confirmed by SDS-PAGE analysis of the eluted fractions and listed in a table. Wild-type rKAL (blue line), rKAL-K307/308D (green line) and rKAL-K312/313D (red line) were eluted from the column at about 300, 95, and 125 mM NaCl concentration, respectively. R259D, R308D, and R394D were eluted from the column at around 160 mM NaCl concentration.

**Figure 4 biomolecules-10-00828-f004:**
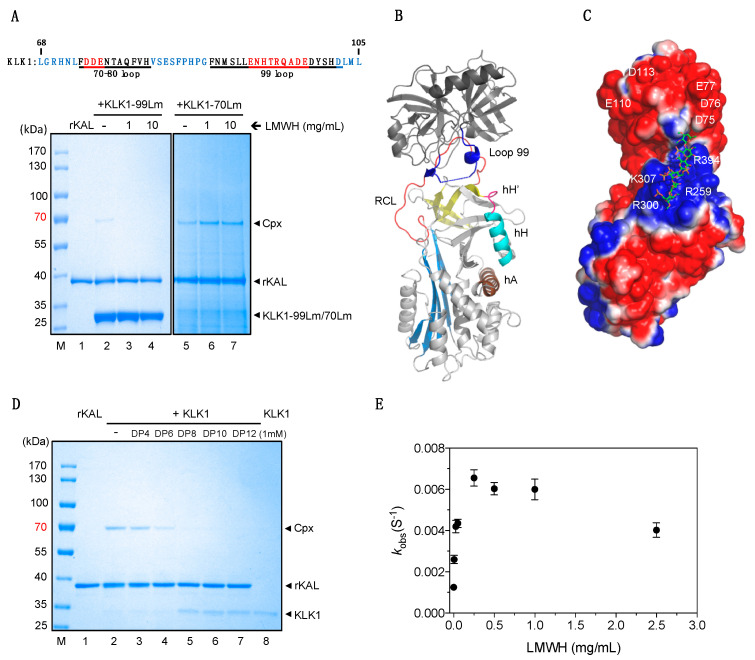
Effect of heparin on the interaction between KLK1 variants and kallistatin. (**A**) Amino acid sequence showing the KLK1 70–80 loop and 99 loop (kallikrein loop) with the negatively charged residues of 70–80 loop highlighted in red. A total of 2 µg of rKAL were mixed with 5 µg of KLK1-99Lm (nine residues highlighted in red from KLK1’s 99 loop were deleted) for 20 min or 0.5 µg of KLK1-70Lm (residues D75, D76 and E77 replaced by arginines) for 2 min in the absence or presence of LMWH at room temperature. (**B**) A model of the KLK1 and kallistatin Michaelis complex was prepared from the thrombin/antithrombin complex structure (PDB 1SR5) with the reactive loop of kallistatin modeled in the active site of KLK1. The long 99 loop of KLK1 is colored blue. (**C**) Electrostatic presentation of the surface charges of the KLK1 and kallistatin Michaelis complex. A heparin chain with six saccharides was modeled along a groove involving R259, K307, R308, and R394 for binding. Further extension of this oligosaccharide would reach the negatively charged surface of KLK1. The electrostatic surface potential was calculated by APBS (red is negative and blue is positive, −1 to +1 kT/e). (**D**) Heparin length effect on the blocking of KLK1 inhibition by kallistatin. A total of 2 µg of rKAL were incubated with 0.5 µg KLK1 protease for 3 min at room temperature in 20 µL PBS, in the absence or presence 1 mM DP4, DP6, DP8, DP10, DP12. All samples were analyzed by reducing SDS-PAGE and stained by Coomassie Blue. (**E**) Effect of LMWH on the inhibition of KLK-70Lm by kallistatin. The observed inhibition rate constants, *k*_obs_, of KLK-70Lm by kallistatin in the presence of different concentrations of LMWH were plotted.

**Figure 5 biomolecules-10-00828-f005:**
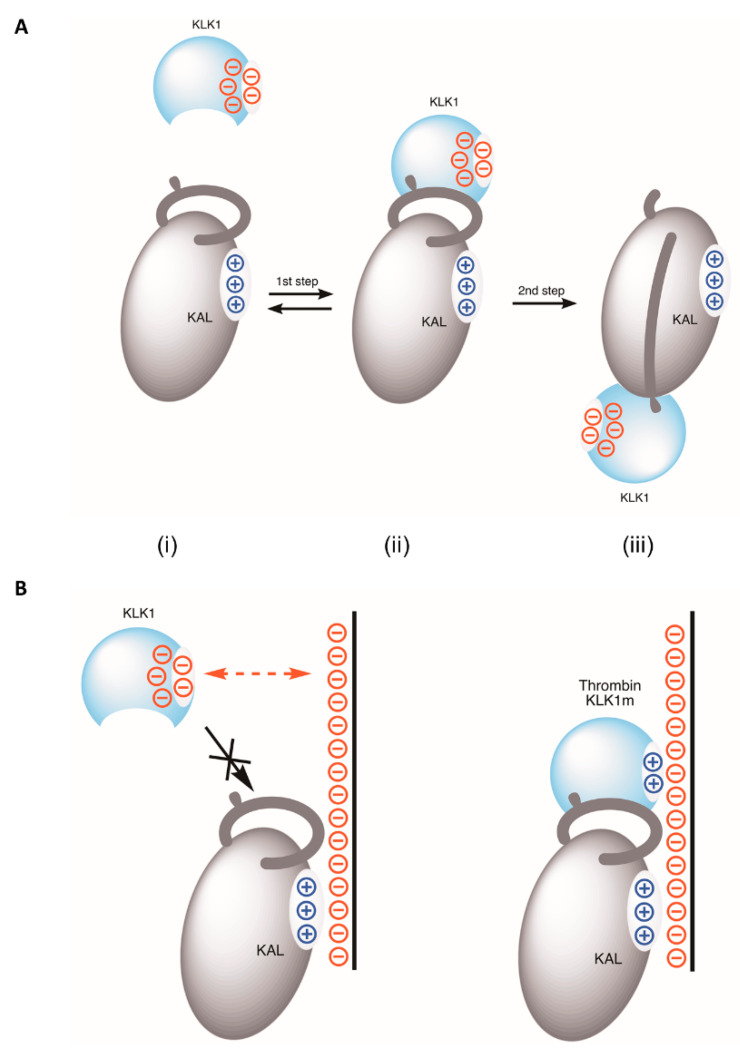
Heparin blocks the inhibition of KLK1 through electrostatic repulsion. (**A**) Kallistatin inhibits KLK1 through the classical two-step serpin inhibition mechanism. Kallistatin (i) in solution or in circulation can form an initial encounter or Michaelis complex with KLK1 (ii). In general, this step is reversible and often subject to modulations by cofactors as seen with many serpins. Subsequently the reactive center loop of kallistatin is cleaved by KLK1 and kallistatin undergoes profound conformational changes leading to an SDS-stable KAL-KLK1 complex linked through an ester bond (iii). This second step of the reaction is irreversible. (**B**) The specificity or activity of kallistatin is regulated by heparin through affecting its Michaelis complex formation. KLK1 is prevented from docking onto kallistatin when kallistatin is bound on heparin or heparinoids in endothelium due to electrostatic repulsion between KLK1 and the negatively charged heparin-like molecules. In contrast, heparin promotes the Michaelis complex formation between kallistatin and proteases such as KLK1 mutant (KLK1m) and thrombin with exosite 1 through the typical bridging mechanism as seen with heparin mediated antithrombin-thrombin interactions. This leads to an accelerated protease inhibition by the serpin.

**Table 1 biomolecules-10-00828-t001:** Data processing, refinement, and model statistics.

Crystal	PDB 6F4U	PDB 6F4V
	Kallistatin	Kallistatin-heparin
Space group	P6_1_	P6_1_
Cell dimensions		
a, b, c (Å)	113.51, 113.51, 76.17	113.77, 113.77, 76.56
α, β, γ (°)	90.00, 90.00, 120.00	90.00, 90.00, 120.00
Solvent content (%)	44	44
Data processing statistics		
Wavelength (Å)	1.196	1.196
Resolution (Å)	35.67–1.9; 2.0–1.9	56.88–1.80; 1.84–1.8
Total reflections	185,915; 26,532	366,297; 13,230
Unique reflections	44,414; 6433	52,290; 3094
Multiplicity	4.2; 4.1	7.0; 4.3
Mean I/sd(I)	14; 2.5	15.3; 1.5
Completeness (%)	99.9; 99.7	99.9; 99.3
R_merge_	0.043; 0.326	0.061; 0.648
R_meas_	0.049; 0.374	0.071; 0.832
Model		
Number of atoms modeled	3293	3364
Protein	3058	3067
Water	185	230
Refinement statistics (Å)		
Reflections in working/free set	42,206/2208	49,688/2602
R-factor/R-free (%)	0.168/0.193	0.173/0.196
^a^ r.m.s. deviation of bonds (Å)/angles (°) from ideality	0.0068/1.12	0.0114/1.46
Ramachandran plot (favored/outliers, %)	98.40/0	95.89/0
Wilson B-factor (Å^2^)	30.1	28.1
MolProbity score	0.98, ^b^ 100th percentile (*n* = 12,147, 1.9 ± 0.25 Å)	1.04, ^b^ 100th percentile (*n* = 11,444, 1.80 ± 0.25 Å)

^a^ r.m.s., root mean square; PDB, Protein Data Bank. ^b^ 100th percentile is the best among structures of comparable resolution; 0th percentile is the worst.

**Table 2 biomolecules-10-00828-t002:** Second order association rate constants for the interaction between kallistatin and tissue kallikrein 1 (KLK1). The apparent second order rate constant (*k_app_*) was determined from the slope of the linear plot of *k_obs_* versus the inhibitor concentration. KLK1 was mixed with different amounts of kallistatin and linear regression analysis of the decrease in protease activity with the concentration of kallistatin yielded the stoichiometry of inhibition (SI) as the intercept on the abscissa. The product of *k_app_* × SI represents the second order rate constant (*k*_2_). Each experiment was repeated three times with mean ± SD calculated.

Protein	*k*_app_ (*M*^−1^*s*^−1^)	SI	*k*_app_ × SI (*M*^−1^*s*^−1^)
rKAL-WT	2.4 ± 0.1 × 10^3^	1.4 ± 0.1	3.4 × 10^3^
rKAL-R259D	2.2 ± 0. 1 × 10^3^	1.6 ± 0.1	3.5 × 10^3^
rKAL-R300D	1.6 ± 0.1 × 10^3^	1.8 ± 0.1	2.9 × 10^3^
rKAL-R394D	1.0 ± 0.1 × 10^3^	1.8 ± 0.1	1.8 × 10^3^
rKAL-307/308D	1.3 ± 0.1 × 10^3^	1.5 ± 0.1	2.0 × 10^3^
rKAL-312/313D	1.0 ± 0.1 × 10^3^	1.5 ± 0.1	1.5 × 10^3^

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
