# Peer review of "Heparin Blocks the Inhibition of Tissue Kallikrein 1 by Kallistatin through Electrostatic Repulsion"

_biomolecules, 2020, doi:10.3390/biom10060828_

Round 1
Reviewer 1 Report
The authors analyze heparin binding on the interaction of the serpin kallistatin with the protease kallikrein 1. The findings are interesting and with significant interest for the community. The presentation and discussion is overall clear and sound. The language needs minor editing, the native English speaker(s) should take care of this. The following points have to be addressed.
The crystallographic analysis and presentation of these results appears to have been done with care and professionally. The structures reported here are already deposited in the PDB since Nov 2017 and are available for download. This is fine. However, the authors have to describe very clearly that heparin binds directly at a crystal contact and that heparin interacts with positively charged residues (Lys-150) AND VIA additional hydrogen bonding interactions with a neighboring molecule in the crystal. The physiological relevance of the detailed binding mode observed here is therefore unclear. Further it should mentioned that the ligand binding site is only partially occupied with an occupancy of 0.69. Due to the partial occupancy and the somewhat higher B-factors of the two heparin units, the electron density is somewhat weak, which is expected. With the map coefficients available from the PDB I have to use lower contour levels than 1.0 rmsd to get an electron density map as depicted in Figure 2B, please check. The authors should calculate a phenix polder map for the ligand.
Page 8, figure 2, panels E and F: The electrostatic potential at the molecular surface looks unusual in that surface areas with distinct positive potential are directly next to areas with distinct and strong negative potential without regions of low potential (white or weakly colored) in between. In addition, there are many small areas which have opposite potential as the surrounding area, again with no border of potential close to zero. This should not be the case and something may have gone wrong. Please check. This is also the case for Figure 4C. State the magnitude of electrostatic potentials in kT/e used for coloring. Explain the coloring in panels C, D, and G. Again, omit description of results but explain what is shown.
At several places in the text the authors phrase "positively charged surface". Strictly speaking, the molecular surface is not charged, but it has positive or negative electrostatic potential caused by full and partial charges located at residues or atoms.
The concentrations of heparin used to block KLK1 inhibition by rKAL in F1A or F4A are very high (up to 2mM). Does heparin affect KLK1 activity?
The authors speculate on the physiological relevance of heparin binding of KAL and a potential localized inactivation on the cell-surface of endothelial cells. To better estimate physiological relevance and likeliness of KAL binding to ECM GAGs, dissociation constants of rKAL and double mutants (K307/R308 and K312/313D) should be determined for DP4, Dp10 and LMWH at an ionic strength of 0.15 M NaCl, e.g. by TNS fluorescence titration as previously done by the authors for lamprey angiotensinogen (Wei et al. JBC 2016). While Kd determinations for similar rKAL alanine-mutants have been done before by Chen et al. (JBC 2001), a detailed analysis with the variants characterized here would help also to decipher the role of K312/313. These are obviously crucial residues in functioning both as heparin binding residues (highest loss in affinity in chromatography and in Kd from Chen et al.) and exosite residues (negligible inhibition of KLK1-99Lm).
Did the truncation/mutation of kallikrein loops in KLK1-99Lm and 70Lm affect protease activity? This should be made clear, as this would additionally affect the inhibition rate of KLK1-99Lm by rKAL.
The authors demonstrate that the KLK1-70Lm variant is inhibited by rKAL and that heparin, due to the lack of repulsion at the mutated KLK1 70-80 loop, does accelerate the inhibition rate. This is evident from the SDS-gel. They further present the measured inhibition rate of rKAL for KLK1-70Lm as 1.8x103 M-1s-1 and mention that this could be accelerated ~5-fold in a heparin dose-dependent manner. These experiments are not described in the methods section and the results. Exact kapp and SI in the presence of heparin together with the bell-shaped acceleration should be presented. Which concentration of heparin yielded the highest acceleration? In the SDS-gel of F4A an increase in complex formation is observed in the presence of up to 10 mg/ml (2mM) of LMWH. What is the concentration of serpin and how many-fold is the resulting excess of heparin in these assays? With respect to the bell-shaped dose-response, which concentrations did finally counteract the template/bridging effect?
KAL, in addition to KLK1, also inhibits further KLK proteases in KLK7 and KLK14. This brings to mind a different serpin, vaspin/SERPINA12, that also inhibits these two KLK proteases and where heparin accelerates inhibition of KLK7, but prevents inhibition of KLK14. It would be interesting to speculate on how the presented findings translate to other protease targets of KAL such as KLK7 or KLK14.
Minor points
page 3, line 117 and line 119: "inclusion body" -> "inclusion bodies"
page 3, line 141: searching model -> search model
page 5, line 188: "low molecular heparin (LMWH)" -> "low molecular weight heparin (LMWH)". Rephrase "relatively little complex could be seen from the gel"
page 6, line 198, The figure legend is much too long. A figure legend should usually only contain the information necessary to understand what is depicted in the figure, but no descriptions or interpretations of results, these should go into the main text.
Reviewer 2 Report
This is a great article from the Aiwu Zhou lab, which defines the heparin binding site in kallistatin using structural analysis and mutagenesis studies, and explains the effect on kallikrein inhibition.
I have only a few doubts that I would like the authors to try to comment. As kallistatin is a glycoprotein, the recombinant expression in E. coli does not allow studying the effect of glycosylation on heparin binding. Could the authors speculate on the possible effect of glycosylation on the binding of heparin to kallistatin?
On the other hand, could KLK1 expression without the propeptide affect the folding of the protein?
Although the effect of heparin on KLK1 inhibition by kallistatin was not due to steric impairment of exositic interactions between KLK1 and kallistatin, could glycosylation of kallikrein affect the binding of heparin and thus the inhibition?
Based on the structure that the authors have solved, what are the differences in KLK1 inhibition in the presence of LMWH and HMWH?
Round 2
Reviewer 1 Report
Line 556: ploder map -> polder map
Figure legend Figure S2 in first and second line: ploder map -> polder map
Line 325, Figure legend 2(G): "The positions of helix G and H are slight differences between them, showing an overall similar configuration with slight differences in the positions of helix G and H."
Please rephrase this sentence, suggestion: "Helices G and H have slightly different positions and orientations."
Author Response
Dear editor,
We thank reviewers for their professionalism.
We have now corrected two typos as pointed out by this reviewer.
(Line 270 and in Figure S2 legend)
Also we have re-phrased the sentence in line 303 as suggested. "Helices G and H have slightly different positions and orientations."
Furthermore, the body of the text has been further and carefully checked by the English co-author for clarity of usage and some small additional corrections in grammar have been incorporated.
The detailed changes are highlighted yellow in a 'clean' version of the manuscript downloaded from the submission system.
We would like to thank your help in getting a speedy turnaround of the manuscript. It IS so impressive, especially during this pandemic time!
If you require further information, please let me know.
Best regards,
Aiwu